# Distinct cortical systems reinstate the content and context of episodic memories

James E. Kragel [1✉], Youssef Ezzyat[1], Bradley C. Lega [2], Michael R. Sperling [3], Gregory A. Worrell [4], Robert E. Gross[5], Barbara C. Jobst[6], Sameer A. Sheth[7], Kareem A. Zaghloul [8], Joel M. Stein[9] & Michael J. Kahana[1✉]

Episodic recall depends upon the reinstatement of cortical activity present during the formation of a memory. Evidence from functional neuroimaging and invasive recordings in humans suggest that reinstatement organizes our memories by time or content, yet the neural systems involved in reinstating these unique types of information remain unclear. Here, combining computational modeling and intracranial recordings from 69 epilepsy patients, we show that two cortical systems uniquely reinstate the semantic content and temporal context of previously studied items during free recall. Examining either the posterior medial or anterior temporal networks, we find that forward encoding models trained on the brain's response to the temporal and semantic attributes of items can predict the serial position and semantic category of unseen items. During memory recall, these models uniquely link reinstatement of temporal context and semantic content to these posterior and anterior networks, respectively. These findings demonstrate how specialized cortical systems enable the human brain to target specific memories.

---

[1] Department of Psychology, University of Pennsylvania, Philadelphia, PA, USA. [2] Department of Neurosurgery, University of Texas Southwestern, Dallas, TX, USA. [3] Department of Neurology, Thomas Jefferson University, Philadelphia, PA, USA. [4] Department of Neurology, Mayo Clinic, Rochester, MN, USA. [5] Department of Neurosurgery, Emory School of Medicine, Atlanta, GA, USA. [6] Department of Neurology, Dartmouth-Hitchcock Medical Center, Lebanon, NH, USA. [7] Department of Neurosurgery, Columbia University Medical Center, New York, NY, USA. [8] Surgical Neurology Branch, National Institutes of Health, Bethesda, MD, USA. [9] Department of Radiology, Hospital of the University of Pennsylvania, Philadelphia, PA, USA. ✉email: james.kragel@northwestern.edu; kahana@psych.upenn.edu

Episodic recall allows us to remember the past, bringing back memories from a specific place or time. This type of memory retrieval involves the reinstatement of encoding-related neuronal activity that codes for memory attributes[1] (e.g., a specific person[2] or place[3]). Neural reinstatement has been proposed as a mechanism for targeting individual memories during memory search[4,5], and subsequent studies have demonstrated content[2,6,7] and context[3,8] reinstatement preceding memory recall. Thus, these two types of reinstatement may serve to target memories that contain certain content or are placed in specific contexts. However, because studies typically focus on either content or context reinstatement in isolation, we know relatively little about how these types of information reinstate across the brain.

Consistent with the longstanding distinction between episodic and semantic memory[9] neuroimaging and electrophysiological studies suggest that separable cortical systems support memory for the semantic meaning and the spatiotemporal context of experienced events[10–14]. Specifically, researchers have identified a posterior medial (PM) network of regions, including parahippocampal, retrosplenial, and posterior parietal cortices, that activate during the processing of contextual information[15,16]. By contrast, other work has identified an anterior temporal (AT) network of regions that appears critical for semantic and conceptual memory[17,18]. This network includes the ventral temporal pole and perirhinal cortices[19]. Because these two networks interact with the hippocampus during memory formation and retrieval[14,20], we predicted reinstatement within these two systems would reflect either the content or context of retrieved memories.

Neuroimaging studies that measure population-level neuronal activity have demonstrated reinstatement across cortical systems during memory retrieval. For example, content-related patterns of spectral power from intracranial electroencephalography (iEEG) across prefrontal and temporal cortices reinstate prior to memory recall[6,21]. Slowly changing patterns of iEEG power in the temporal lobe, consistent with a representation of temporal context[5], reinstate during retrieval and account for temporal organization of recall sequences[8]. Functional MRI studies, which provide greater sampling of cortical systems than invasive recording techniques, have demonstrated contextual reinstatement within the hippocampus[22] and functionally connected PM regions[23]. Multiple regions within the PM network, including the angular gyrus and medial prefrontal cortex, reinstate event-specific patterns of activity[24], suggesting this network may be involved in representing the content of memories within a specific episodic context. It is possible that these findings stem from representations of content and context integrated within a single cortical system, coding the full range of attributes in memory. In contrast, the PM and AT networks may independently drive hippocampal-dependent retrieval, serving as distinct cortical pathways to recall.

To examine the contributions of cortico-hippocampal networks to recall behavior, we utilized a computational modeling technique originally developed to predict patterns of brain activity based on the semantic content of individual stimuli[25–27]. This method takes advantage of the sensitivity of neural signals to semantic attributes of presented items. By learning how activity in the brain is shaped by semantic attributes, these models can reliably decode the semantic content of stimuli from an observed pattern of brain activity. We extended this technique to develop context-based models that were trained to predict patterns of brain activity based on the temporal context (i.e., at which point in time in the experiment) in which stimuli were presented. By applying these models to iEEG signals recorded while subjects performed a free-recall task, we tested whether content- and context-based memory representations are reinstated within distinct cortical systems: the AT and PM networks.

## Results

We recorded iEEG from neurosurgical patients ($n = 69$) while they performed a free-recall task with items drawn from 25 distinct categories presented in same-category pairs (Fig. 1a). Patients recalled an average of 31% (0.02 SEM) of list items. Both the serial position and category of items influenced their probability of recall. We observed a primacy effect, as evidenced by a recall advantage for items presented in the first (mean 49%, 0.03 SEM; $t_{68} = 9.2$, $p < 0.0001$) and second (mean 38%, 0.03 SEM; $t_{68} = 4.9$, $p < 0.0001$) serial positions (Fig. 1b). To evaluate category-level differences in recall performance, we compared the proportion of items recalled from each category to the average across all items (Fig. 1c). We found patients had better memory for zoo animals (mean 42%, 0.03 SEM; $t_{68} = 4.1$, $p = 0.0001$) and weather (mean 39%, 0.02 SEM; $t_{68} = 3.3$, $p = 0.001$), and worse memory for electronics (mean 22%, 0.02 SEM; $t_{68} = -4.4$, $p < 0.0001$).

In addition to overall recall performance, the serial position and category of items influenced the order in which items were recalled. Recall clustering (e.g., consecutively recalling same-category items) indicates that the retrieval cue used during memory search targets certain properties of the studied items. Because no retrieval cues are provided to the subject during free recall, the cue must be internally generated by the reinstatement of information present during the study period. To examine category clustering, we used a word embedding model[28] to derive a vector representation of each studied item. To illustrate the similarity structure derived from the word2vec model, we projected the 300-dimensional word representations onto a three-dimensional space derived using principal component analysis (Fig. 1a, right). We measured clustering with temporal and categorical factor scores, which indicate whether clustering is above or below chance levels[29] (see Fig. 1d for example measures). Recall sequences exhibited both categorical ($t_{68} = 30.7$, $p < 0.0001$) and temporal ($t_{68} = 10.8$, $p < 0.0001$) clustering (Fig. 1e). These clustering effects remained after accounting for potential confounds due to the list structure (see Methods). Adjusted measures of categorical ($t_{68} = 25.6$, $p < 0.0001$) and temporal ($t_{68} = 11.4$, $p < 0.0001$) clustering also indicated significant levels of recall organization (Supplementary Fig. 2). Both recall performance and clustering demonstrated reinstatement of both semantic content and temporal context occurred during memory search.

**PM and AT networks contain representations of context and content during encoding.** To assess representations of temporal context and semantic content within PM and AT networks, we first defined these networks based on an independent resting-state fMRI analysis (Fig. 2a; see Methods for details)[30]. This analysis follows the standard method of correlating the BOLD time series between a region of interest and other regions distributed throughout the rest of the brain[31]. Regions with connectivity to temporal polar cortex defined the AT network. These regions included inferior lateral prefrontal cortex, anterior temporal cortex, and inferior angular gyrus (Fig. 2a, green). Regions with connectivity to the posterior angular gyrus defined the PM network (Fig. 2a, purple). Cortical regions within this network included the posterior cingulate cortex, precuneus, parahippocampal cortex, and posterior parietal cortex (see Supplementary Fig. 1 for details regarding electrode coverage).

Synchronous low-frequency activity across the brain signals memory processing[32], with theta oscillations responsible for interactions between hippocampus and cortex during memory search[33]. Prominent theories propose synchronous interactions coordinate network communication[34], which would support hippocampal-dependent reinstatement of memory content. Thus,

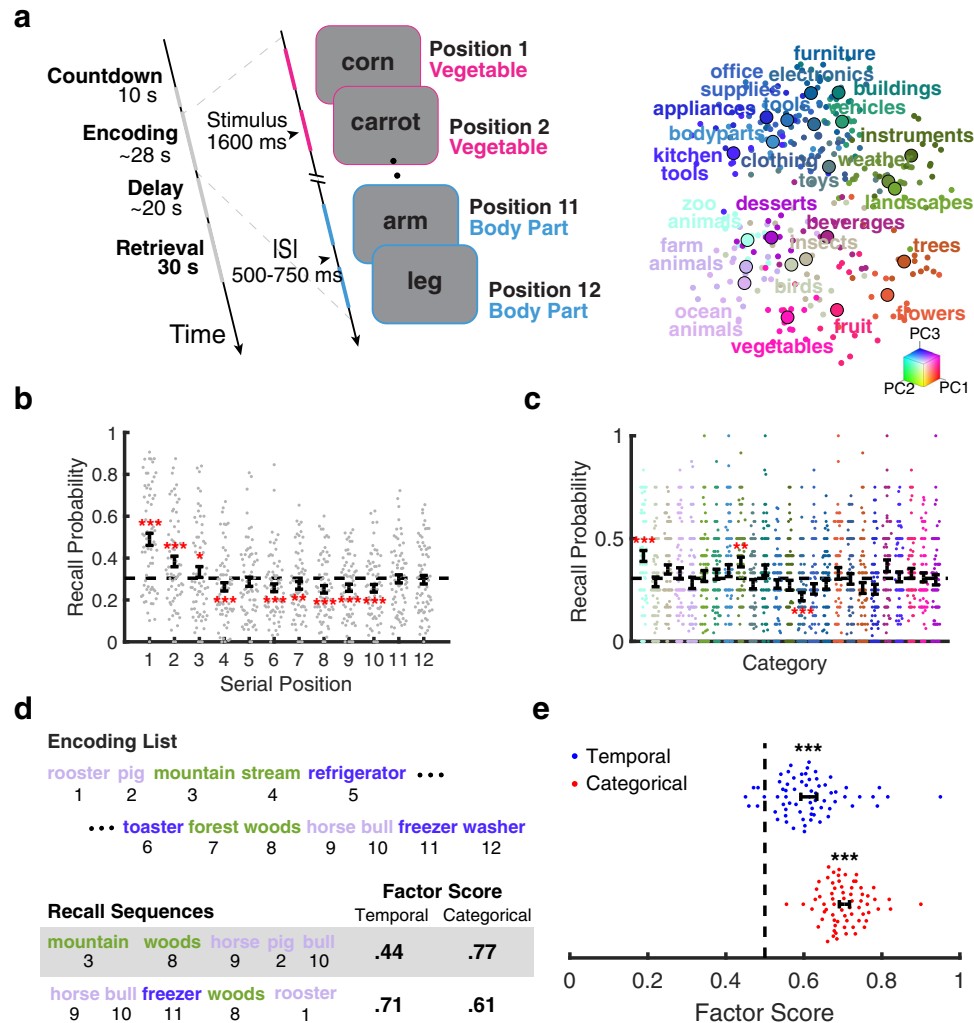

**Fig. 1 Task and behavioral results. a** Schematic of the free-recall task. Left: Task timeline for a single recall trial, highlighting the list structure and timing during the encoding period. Subjects completed 25 lists per experimental session. Right: Category representations of stimuli. Each point indicates a word's semantic representation along with the first three principal components of the word2vec[28] space. Larger points denote the midpoint of all stimuli within a category. Words with similar meaning are spatially proximal. **b** Recall performance by serial position. Compared to average recall (dashed line), subjects ($n$ = 69, indicated by individual points) recalled significantly more items from early serial positions. **c** Recall performance by category. Plotting convention follows panel (**b**). **d** Factor scores for example recall sequences. Factor scores above 0.5 indicate above chance levels of temporal or categorical organization of recall sequences. **e** Recall organization. Subjects ($n$ = 69, indicated by individual points) clustered recalls along temporal and semantic dimensions, as indicated by above chance (dashed line) factor scores. Statistical inference was performed with two-sided, one-sample $t$ tests versus chance (**e**) or average recall performance, with FDR correction for multiple comparisons (**b**, **c**). ***$p < 0.001$, **$p < 0.01$, *$p < 0.05$. Data are presented as mean values ± SEM. Source data are provided as a Source Data file.

we predicted synchronous activity in both PM and AT networks during the free-recall task. We next tested the hypothesis that PM and AT networks defined by resting-state fMRI would predict correlations in low-frequency power during task performance. Previous work has established correspondence between networks defined by resting-state fMRI and iEEG[35,36]. However, it is possible that different network structures emerged in iEEG signals during the recall task. For example, these two cortical systems could reconfigure into a single network. We predicted distinct networks to emerge as subjects perform the task, supporting the hypothesis that separable cortical systems are involved in the representation of content and context.

To examine the separability of PM and AT networks, we asked whether theta exhibited greater connectivity within than between the two networks. We measured connectivity by correlating spectral power across trials of the free-recall task. Such trial-by-trial variability in power reflects endogenous fluctuations across

neural networks[35,37,38]. Because these estimates of connectivity can be spuriously elevated due to proximity of recording sites[36] (see Supplementary Fig. 3 for evidence of drop off in connectivity with inter-electrode distance), we used a bootstrap matching procedure to control for differences in distance between contacts located within the same network or across both networks (see Methods). Using this approach, low-frequency power was significantly more correlated within the AT network than between the PM and AT network. This effect held at the lower (3 Hz, $t_{64} = 4.97$, $p < 0.0001$), intermediate (5 Hz, $t_{64} = 3.67$, $p < 0.0001$), and upper range of theta (10 Hz, $t_{64} = 3.12$, $p < 0.0001$). Within the PM network, we found evidence for synchronous activity near the upper end of theta (10 Hz, $t(64) = 2.13$, $p = 0.04$). We also explored whether network-specific coupling occurred at higher frequencies, spanning beta and gamma ranges (frequencies surviving FDR corrected thresholds are highlighted in Fig. 2a, bottom). We did not identify any higher frequency effects within

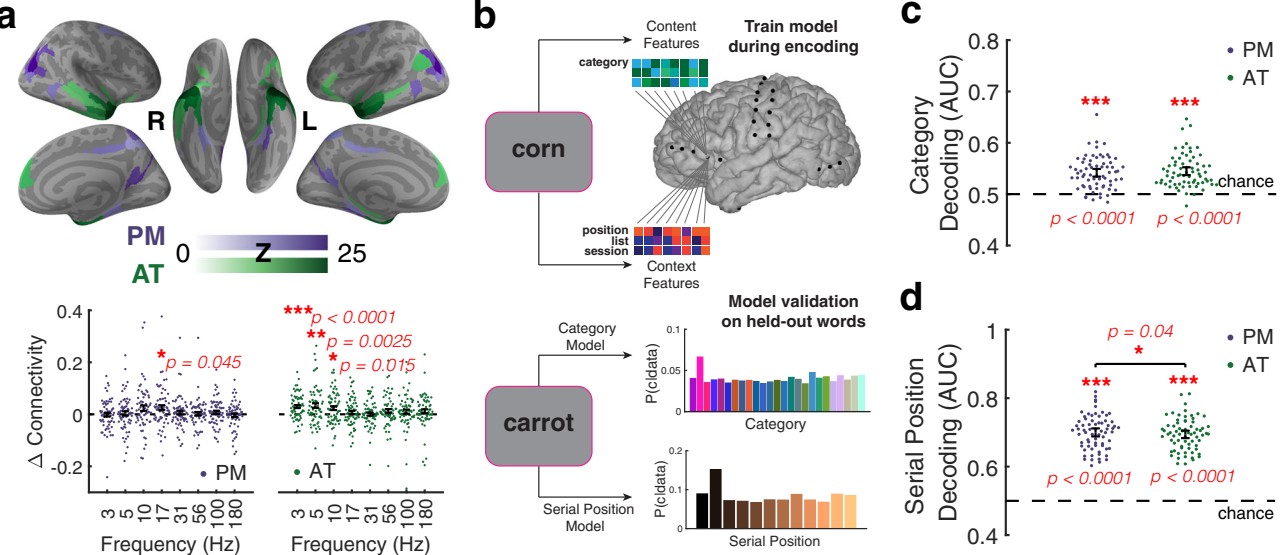

**Fig. 2 Network-based decoding of content and context. a** rsfMRI connectivity identifies distinct cortical networks. Top: Functional connectivity (Fisher Z) to cortical targets of the PM and AT networks across right (R) and left (L) hemispheres. Bottom: Difference in connectivity of recording sites within versus between the two networks. Frequency specificity was observed within each network. **b** Modeling schematic. Ridge regression modeled evoked neural activity during stimulus encoding. Models were validated by decoding the category and serial position of held-out stimuli during encoding. **c** Reliable category decoding in the PM and AT networks during encoding. **d** Accurate serial position decoding for the PM and AT networks during encoding. Statistical inference was performed with two-sided, permutation $t$ tests versus zero with FWER correction for multiple comparisons (**a**) or with one-sided sign tests versus chance (**c, d**). For all tests, $n = 69$ subjects. Data are presented as mean values ± SEM. $*p < 0.05$, $**p < 0.01$, $***p < 0.001$. Source data are provided as a Source Data file.

the AT network. Coupling in the low beta range was enhanced within the PM network (17 Hz, $t_{64} = 2.70$, $p = 0.009$). These results show that changes in spectral power were coherent within each of these two networks, consistent with their identification as dissociable networks from resting-state fMRI.

Having validated the PM and AT networks in our iEEG recordings, we next sought to evaluate their respective roles in the reinstatement of memories' semantic content and temporal context. To do this, we trained multivariate models to predict brain activity based on the semantic content and temporal context of items presented at encoding. We modeled changes in spectral power from 3 to 180 Hz at each recording site from either the content- or context-based attributes of each item (Fig. 2b). Using these models, we predicted the pattern of spectral power for each serial position or category. We compared these predictions to held-out items in a cross-validation procedure that tested whether it was possible to decode either the serial position or category from brain activity observed within a given network. In addition, applying these models to patterns of spectral power during memory search allowed us to compare reinstatement across the two networks.

We assessed each model's decoding performance during the encoding phase of the experiment, as subjects studied list items. We used a bootstrap procedure ($n = 1000$) to estimate decoding performance from five randomly sampled recording sites within each network (this approach allowed us to control for greater electrode coverage in the AT network, see Supplementary Fig. 1b and Methods for further details). Applying this technique, we reliably decoded the category of items in both the PM (median AUC = 0.54, $Z = 6.9$, $p < 0.0001$) and AT (median AUC = 0.54, $Z = 7.1$, $p < 0.0001$) networks (Fig. 2c), and category decoding did not differ between the two networks (median $\Delta$AUC = 0.002, $Z = 0.8$, $p = 0.45$). We also reliably decoded the serial position of items in both the PM (median AUC = 0.71, $Z = 7.2$, $p < 0.0001$) and AT (median AUC = 0.70, $Z = 7.2$, $p < 0.0001$) networks

(Fig. 2d). Here, our serial position model achieved more accurate decoding in the PM than in the AT network (median $\Delta$AUC = 0.002, $Z = 2.1$, $p = 0.04$). These bootstrap results provide a lower bound for the true decoding performance for a given network (see Supplementary Fig. 4) and confirm the representation of semantic and temporal information across the PM and AT networks during encoding.

**Reinstatement of content and context in PM and AT networks.** Having established the ability to decode temporal context (serial position) and semantic content (category) from encoding-period neural activity in the PM and AT networks, we next asked whether these two types of information reinstated during free recall. To measure reinstatement, we tested our category and serial position models on patterns of neural activity in the moments preceding recall (from 900 to 100 msec prior to vocalization, in 20 msec intervals; Fig. 3a). We observed reinstatement of temporal information in the PM network and semantic information in both the PM and AT networks. Temporal context reinstated in the PM network from 600 to 400 msec before overt recall (Fig. 3b, top panel). Content reinstatement was sustained throughout the pre-recall period (Fig. 3b, bottom panel) in both PM and AT networks. Moreover, these two forms of reinstatement differed between the two networks, with greater content reinstatement in the AT network (Fig. 3b; $p < 0.05$, FWER corrected). These findings demonstrate the specificity of contextual reinstatement within the PM network. In contrast to previous studies that relied upon similarity in neural activity over time to identify contextual information[8,22,23,39,40], we demonstrate the ability to recover the serial position of recalled items from reinstatement in the PM network. We also show greater, sustained content reinstatement within the AT network. These unique patterns of reinstatement reveal a dissociation between content and context reinstatement across the PM and AT networks.

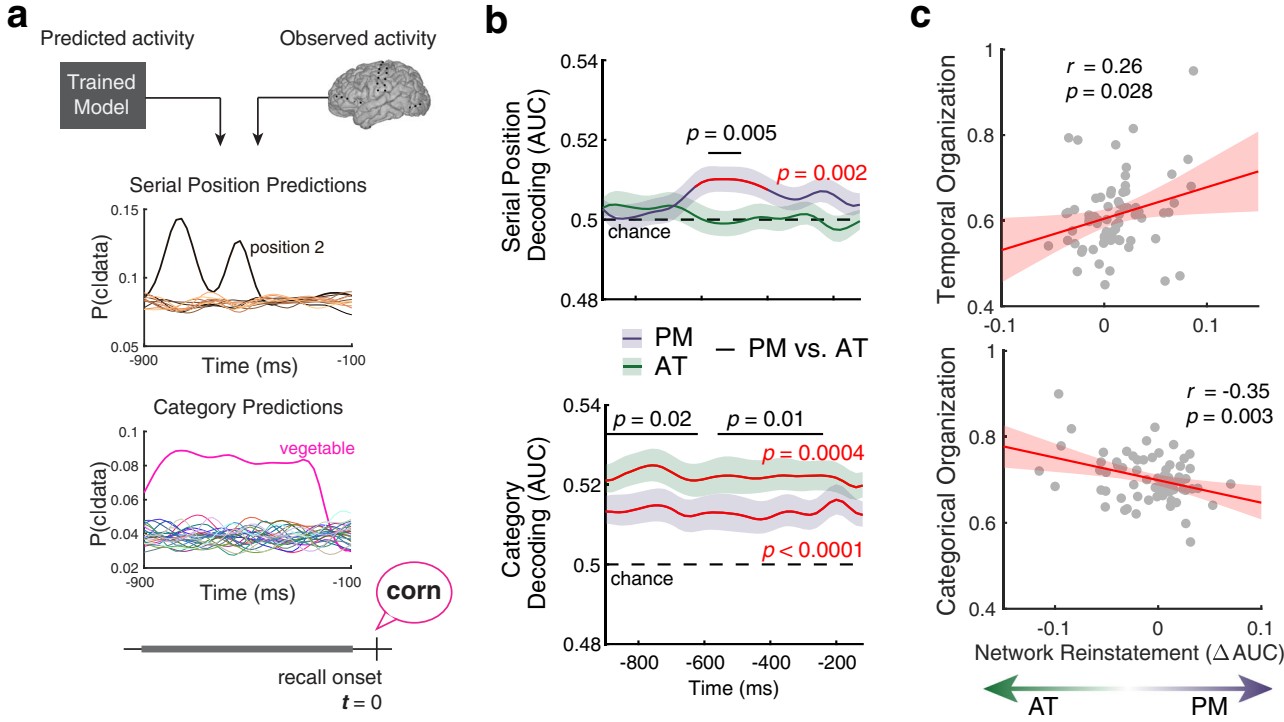

**Fig. 3 Reinstatement of content and context information during memory search. a** Simulated reinstatement analysis. Models trained during encoding predicted the serial position (top) or category (bottom) of the upcoming recall. **b** Decoding performance from reinstated patterns of neural activity within each network. Top, the serial position model revealed transient context reinstatement within the PM network. Bottom, the category decoding model revealed greater content reinstatement within the AT than the PM network. Significant differences in reinstatement between the two networks are indicated in black. Red lines highlight timepoints with above chance decoding, $p < 0.05$, FWER corrected (two-sided permutation $t$ test). **c** Relationship between network reinstatement and organization of recall sequences along temporal and categorical dimensions. Each point denotes an individual subject. Top, greater context reinstatement in the PM network was associated with increased temporal organization. Bottom, greater content reinstatement in the AT network was associated with greater category organization. Uncorrected $p$ values from two-sided Student's $t$ tests are displayed (**b**, **c**), after averaging over significant timepoints (FWER corrected, **b**). For all tests, $n = 69$ subjects (indicated by points in **c**). Shaded regions denote mean ± 1 SEM. Source data are provided as a Source Data file.

Our finding that reinstatement of temporal context is specific to the PM network relies upon chance-level decoding within the AT network. It is possible that by limiting the number of electrodes included in this analysis (which was necessary for comparisons between networks), we impaired our ability to detect reinstatement, particularly if the information was represented in a distributed manner. To evaluate this possibility, we examined decoding performance after incorporating additional electrodes into our analysis (Supplementary Fig. 4). Adding electrodes produced more robust category reinstatement effects in both the PM ($\chi_1^2 = 5.7$, $p = 0.02$) and AT ($\chi_1^2 = 31.4$, $p < 0.0001$) networks. This analysis also revealed stronger context reinstatement in the PM ($\chi_1^2 = 6.7$, $p = 0.009$) but not the AT ($\chi_1^2 = 0.8$, $p = 0.36$) network when incorporating additional features. These findings provide additional support for the specificity of context reinstatement to the PM network and suggest that decoding accuracy was limited by anatomical coverage and the spatial resolution of recordings.

To rule out the possibility that epileptiform activity influenced our ability to detect reinstatement, we examined whether electrodes that were either within the seizure onset zone, exhibited inter-ictal spiking, or located near pathological tissue impacted decoding performance. We repeated our decoding analyses after excluding these electrodes and found a remarkably similar pattern of results (Supplementary Fig. 5). We found no evidence that signals from these electrodes impacted our ability to decode information at encoding (all $\chi_1^2 < 0.25$, $p's > 0.61$) or recall (all $\chi_1^2 < 0.43$, $p's > 0.53$). As such, reinstatement reflected changes

in spectral content associated with physiologically and cognitively normal processes.

Cortical reinstatement has been shown to predict how memory search unfolds[4,21], with cortical representations providing a top-down cue for the memory system to retrieve specific information. As such, it is possible that the PM and AT networks are responsible for targeting stored memories with specific content- or context-based attributes. To test this hypothesis, we examined whether differences in reinstatement across the two networks predicted the tendency of subjects to organize their recall sequences along temporal or categorical dimensions. Consistent with prior work linking cortical reinstatement to memory organization[6,8], variability in the organization of a subject's memory was predicted by cortical reinstatement during memory search (Fig. 3c). Greater reinstatement of context-based representations within the PM network tracked the tendency to consecutively recall items from nearby serial positions ($r_{67} = 0.26$, $p = 0.028$). Increased reinstatement within the AT network was associated with greater organization based on the categorical structure of the list ($r_{67} = -0.35$, $p = 0.003$). Representations encoded in these two cortical systems can differentially guide memory search, biasing how and what we remember.

## Discussion
We report evidence that the reinstatement of content- and context-based information occurs within distinct cortical networks. Furthermore, the quality of reinstatement within a given network was predictive of a subject's tendency to organize their

memories along either temporal or categorical dimensions. These results help resolve conflicting viewpoints on how distinct neural representations contribute to memory search. Episodic retrieval occurs when a memory cue converges on the hippocampal formation, prompting associative recall[41]. Multiple cortical structures have been proposed to represent temporal context, which cues episodic recall[42], including lateral prefrontal cortex[5] and parahippocampal inputs to the hippocampus[43]. Despite evidence implicating these structures in representing temporal information[8,22,44–47], it has been unclear whether these brain regions are the primary drivers of episodic recall. Our findings of greater reinstatement of context within the PM network and content within the AT network resolve this ambiguity by characterizing the contributions of these cortico-hippocampal networks to memory search.

Our findings complement previous studies of the neural bases of memory search[6,8,21] that have linked neural reinstatement to the organization of human memory. Along with fMRI investigations of temporal coding in the MTL and cortico-hippocampal networks[22,23], these previous studies identified candidate regions for the representation of temporal context based on slow changes in the similarity of neural patterns over time. Here, our modeling approach enabled us to recover the temporal position of items within each list, irrespective of the list in which the item was encoded. By doing so, we rule out the possibility that context-like signals emerge from the covert retrieval or maintenance of list items throughout study lists, an inherent limitation of prior work. Given the PM network specifically reinstates contextual codes that predict the temporal organization of recall sequences, neuronal interactions within this network support the ability to target memories from specific episodic contexts.

The observed temporal coding in the PM network parallels the firing of hippocampal neurons in rodents[47] and so-called "time cells" in humans[48,49]. These neurons show selective increases in firing at specific temporal positions, providing a basis for sequence learning and recall. Emergence of this hippocampal code likely depends on two types of time-modulated inputs from the entorhinal cortex[50]. Neuronal firing within the lateral entorhinal cortex responds to salient stimuli in the environment and decays at rates sufficient to represent ordinal sequences[51]. The hippocampus also receives inputs from grid cells in the medial entorhinal cortex, which displays coupling to the PM network via parahippocampal cortex[52,53]. These neurons code for an internal representation of continuous time, as they fire at specific temporal intervals during treadmill running[54] and periods of immobility[55] when changes in the environment cannot provide temporal information. Although we did not directly relate time cell activity to cortical reinstatement, our findings suggest temporal coding in the medial entorhinal cortex may be crucial to reinstatement in the PM network.

Because the present findings are inherently correlational in nature, future studies could provide causal evidence between network reinstatement and recall organization via noninvasive or direct electrical stimulation. Adaptation of recently developed closed-loop stimulation techniques[56,57] may provide a framework to modulate hippocampal activity and bias recall of information from a specific context.

Unlike reinstatement of contextual information that was specific to the PM network, we were able to decode the category of recalled items from patterns of activity in both networks. Converging evidence from fMRI studies suggests that regions within this network[58], including the angular gyrus[24], represent recollected content in an episode-specific manner. Our findings suggest that temporal coding may be a unique feature of this network, in contrast to representations coded in the AT network. Understanding the nature of these temporal representations,

including where they originate and how they are integrated with other forms of episodic information remains to be determined.

Our ability to decode the content and context of items in memory reflects multiple neuronal processes. Specifically, the measures of spectral power we examined reflect a combination of broadband and oscillatory processes, including slower (2–4 Hz) and faster (5–10 Hz) theta rhythms that predominate hippocampal networks[59]. Broadband shifts in power indicate when memory processing occurs, including concurrent decreases in low frequency and increases in high-frequency content[60–62]. These broadband shifts are associated with changes in neuronal excitability and correlate with firing rates of locally recorded neurons[63,64]. Thus, increases in neuronal firing within attribute-specific cortical regions would support accurate decoding. At the same time, reduction of oscillations and increased asynchronous activity supports information coding within cortex[65,66]. Changes in the amplitude of narrowband oscillations would therefore contribute to decoding performance, at timescales constrained by the speed of the oscillation. Understanding the relative contributions of oscillatory and broadband signals to memory reinstatement remains an important question for future research.

Neural models of memory search have suggested that an internal representation of context serves as the primary cue for hippocampal-dependent recall[5]. Our findings argue for an alternative account regarding the neural basis of memory search. One possibility is that semantic representations within the AT system can cue memories in a context-independent manner, prompting recall of memories with similar semantic attributes. If content-based representations can guide memory search in this fashion, one would expect individuals to organize their memories based on semantic content, rather than the temporal order in which it was learned. Indeed, subjects who showed greater reinstatement in the AT network exhibited greater semantic and less temporal organization. This hypothesis is further supported by evidence that content-based activity in the inferior temporal cortex can be used as a top-down signal to bias retrieval to targeted memories[7]. Along these lines, the contextual information represented across the PM network focuses retrieval to a specific temporal context. The effects observed here may reflect a general property of memory search, wherein cortical reinstatement serves as a mechanism to target memories based on network-specific representations.

## Methods

**Participants.** Sixty-nine patients (40 male, with an average age of 39 years [SD 12 years]) with medication-resistant epilepsy underwent neurosurgical procedures to implant intracranial electrodes (subdural, depth, or both) to determine epileptogenic regions. On average, the duration of epilepsy at the time of surgery was 19 years (SD 13 years). Seizure onset zones were commonly localized to mesial temporal ($n = 25$), temporal ($n = 22$), prefrontal ($n = 13$), parietal ($n = 28$), and occipital ($n = 7$) cortices. An additional two patients had seizure onset zones within the hippocampus. Seizure onset zones were lateralized to the left ($n = 31$), right ($n = 23$), or both ($n = 11$) hemispheres. Data were collected at Dartmouth-Hitchcock Medical Center (Hanover, NH), Emory University Hospital (Atlanta, Georgia), Hospital of the University of Pennsylvania (Philadelphia, PA), Mayo Clinic (Rochester, MN), Thomas Jefferson University Hospital (Philadelphia, PA), Columbia University Medial Center (New York, NY), and University of Texas Southwestern Medical Center (Dallas, TX). Prior to data collection, the research protocol was approved by the Institutional Review Board at each hospital. Informed written consent was obtained from either the participant or their guardians.

**Free-recall task.** Each subject performed a categorized free-recall task in which they studied a list of words with the intention to commit the items to memory. The task was performed at the bedside on a laptop, using PyEPL software[67]. Analog pulses were sent to available recording channels to enable alignment of experimental events with the recorded iEEG signal. Word presentation lasted for a duration of 1600 ms, followed by a blank inter-stimulus interval (ISI) of 750–1000 ms (see Fig. 1a). Each list contained items from three distinct categories (four items per category), with two same-category items presented consecutively. The total word

pool consisted of 25 distinct categories, with individual items selected as proto-typical items within each category[68]. Presentation of word lists was followed by a 20 s post-encoding delay. Subjects performed an arithmetic task during the delay in order to disrupt memory for end-of-list items. Math problems of the form $A + B + C = ??$ were presented to the participant, with values of $A$, $B$, and $C$ set to random single-digit integers. After the delay, a row of asterisks, accompanied by an 800 Hz auditory tone, was presented for a duration of 300 ms to signal the start of the recall period. Subjects were instructed to recall as many words as possible from the most recent list, in any order during the 30 s recall period. Vocal responses were digitally recorded and parsed offline using Penn TotalRecall (http://memory.psych.upenn.edu/TotalRecall). Subjects performed up to 25 lists in a single recall session.

**Behavioral analysis.** To compute behavioral measures of temporal clustering, we used the temporal factor[29] score. Temporal factor measures the percentile rank of the absolute lag between successive recalls from the full distribution of available lags for items that have yet to be recalled. To measure category clustering, we computed a category factor, which assumes that all items within the same category have a distance of zero and items of different categories have a distance of one. These metrics measure the degree to which recall sequences exhibit organization along temporal or categorical dimensions, with random recall sequences falling at the median of the distribution (i.e., 0.5).

We presented same category exemplars in sequential pairs, as in the sequence $(A_1, A_2, B_1, B_2, C_1, C_2, B_3, B_4, \ldots, A_4)$ where $X_i$ is an exemplar of category $X$. This list structure creates circumstances where random recall sequences produce factor scores that deviate from the expected value of 0.5. Consider the case where a subject recalls four items from a single category in an arbitrary temporal order. After recalling the first item, the subject could either recall the same-category pair (e.g., $A_2$ following $A_1$) or jump to one of the other same-category items ($A_3$ or $A_4$). Because more items are available for recall at long temporal lags, sequences generated with random temporal order produce a lower than expected temporal factor score of 0.45. Measures of category clustering can be similarly biased by the list structure. Consider the case where a subject serially recalls four items starting at an arbitrary position in the list, without regard to semantic information. Because same-category pairs are always recalled consecutively, the expected category factor score for these recall sequences is elevated to 0.64. These examples highlight that factor scores can deviate from 0.5 due to the list structure rather than true recall clustering. To rule out this potential confound, we performed a simulation-based control analysis where we generated null distributions from random recall sequences matched to both the number of items recalled and either the temporal or categorical clustering of the observed sequences. We standardized the observed clustering measures based on these null distributions ($n = 1000$), which indicate the amount of clustering expected by chance given the list structure. This analysis is sensitive to clustering that is not confounded by the list structure, such as temporal clustering across category boundaries and category clustering across large temporal lags.

**Electrophysiological recordings and data processing.** iEEG signal was recorded using subdural grids and strips (contacts spaced 10 mm apart) or depth electrodes (contacts spaced 3–10 mm apart) using recording systems at each clinical site. iEEG systems included DeltaMed & XlTek (Natus), Grass Telefactor, and Nihon-Kohden EEG systems. Signals were sampled at 500, 512, 1000, 1024, or 2000 Hz, depending on clinical site. Preprocessing of iEEG signal was performed with custom Python (v3.3) software. Signals recorded at individual contacts were converted to a bipolar montage by computing the difference in signal between adjacent electrode pairs on each strip, grid, and depth electrode. Bipolar signal was notch filtered at 60 Hz with a fourth-order 2 Hz stop-band Butterworth notch filter in order to remove the effects of line noise on the iEEG signal.

**Anatomical localization.** Anatomical localization of electrode placement was accomplished using independent processing pipelines for depth and surface electrodes. Post-implant CT images were coregistered with presurgical T1 and T2 weighted structural scans using Advanced Normalization Tools[69]. For patients with MTL depth electrodes, hippocampal subfields and MTL cortices were automatically labeled in a pre-implant, T2-weighted MRI using the automatic segmentation of hippocampal subfields multi-atlas segmentation method[70]. Subdural electrodes were localized by reconstructing whole-brain cortical surfaces from pre-implant T1-weighted MRIs using Freesurfer[71], and snapping electrode centroids to the cortical surface using an energy minimization algorithm[72]. Reconstructed surfaces were additionally mapped to a population-average surface[73] that we used to assign network membership based on resting-state connectivity of cortical regions defined by a multi-modal cortical parcellation[30]. PySurfer (https://pysurfer.github.io/) was used to display electrode coverage and statistics on the cortical surface.

**Network assignment and analysis.** We assigned recording sites to networks of interest based on resting-state functional connectivity in an independent set of subjects from the Human Connectome Project[30]. From the Glasser et al. parcel-lation, we assigned parcels with high connectivity to the posterior angular gyrus (area PGp) to the PM network and parcels with high connectivity to the temporal pole (area TGd) to the AT network. The assignment was based on partial

correlations between each parcel and the seed region, to provide a better estimate of direct network connections. Bipolar pairs were assigned to the nearest network if the bipolar centroid was within 8 mm of a parcel within either network, with the exclusion criteria that they could not be within 4 mm of the other network.

To evaluate the properties of these two cortical networks, we used a bootstrap sampling procedure to randomly sample from bipolar pairs within each network. This approach controls for the effect of distance between recording sites on functional connectivity. As the inter-electrode distance is greater between compared to within functional networks (leading to a biased estimate of connectivity within each network), we controlled the distances within each distribution of connections (i.e., within AT, within PM, or between networks). For each subject, we randomly sampled connections between the two networks. Connections within a given network were sampled without replacement to match the distribution between network connections by minimizing the differences in connection length. The functional connectivity within and between each network was computed as the Pearson product–moment correlation across encoding events in the experiment. This procedure was repeated 1000 times, and the intrinsic connectivity at a given frequency was estimated from the bootstrap distribution. Some subjects were excluded from analysis ($n = 4$) in circumstances where electrode coverage prevented well-matched samples across conditions.

A similar bootstrap procedure was used to estimate the decoding accuracy of models trained from each network. Networks with greater electrode coverage are likely to have higher decoding accuracy due to the number of features alone. As a result, we randomly sampled five electrodes from each network prior to estimating the ability to decode content and context information from patterns of brain activity. This sampling procedure was repeated 1000 times per model evaluation, and the average performance across bootstrap distributions was used to indicate model performance for a given subject.

**Spectral power.** To compute spectral power during word encoding, we applied the Morlet wavelet transform (wave number 5) to all bipolar electrode EEG signals across eight logarithmically spaced frequencies from 3 to 180 Hz. Spectral power during recall was estimated from 900 ms to 100 ms preceding the onset of response vocalization for correct recalls. Recall events were required to be free of vocalization onsets in the preceding 1500 ms. Power estimates were log-transformed and down sampled to 50 Hz. To avoid edge artifacts, we included buffers of 1000 ms surrounding events of interest during the computation of spectral power; mirrored buffering was applied to all retrieval-period data. Prior to modeling, all power estimates were standardized using the mean and standard deviation of each session.

**Model fitting and testing.** We used custom MATLAB code (version R2017b; Natick, MA) to construct two models to predict stimulus-related patterns of neural activity as a function of either (1) the taxonomic category of presented stimuli or (2) the serial position of each presented item within the experiment. For the content model, we modeled the neural response as a function of 300 intermediate category features computed using the word2vec model[28] trained from Google News corpora. Category features were computed by averaging semantic representations across all words presented from a given category. This results in the construction of a high-dimensional space that respects the semantic relationships between all of the categories. For the context model, we modeled the neural response to stimuli as a function of serial position within the list, across lists, and across sessions. For subjects in which only a single session was run, the regressor predicting the effect of session on neural activity was excluded.

We fit both the content and context models separately to each neural feature (i.e., spectral power at a given frequency and bipolar pair). Ridge regression was used to identify model parameters that minimized prediction error on the training data, which was randomly assigned using a 5-fold cross-validation procedure, holding out individual words. Within each training fold, we performed an additional 10-fold cross-validation procedure (i.e., nested cross-validation[74]) to optimize the regularization coefficient used in each fold. Across the ten folds, we selected the regularization coefficient (from 50 potential parameters log-spaced from $10^{-2}$ to $10^{10}$) that minimized the mean squared error of the model predictions across the training set. The resulting set of model weights across neural features was used to decode either the serial position or category membership for unseen patterns of brain activity. We computed the probability that the observed brain data belonged to each of the 25 categories (or 12 serial positions) by computing the Pearson product–moment correlation between the observed data and the predicted pattern of brain activity for each model. A softmax function was applied to the resultant evidence for each class, and decoding accuracy was measured using area under the receiver operating characteristic curve (AUC) metric across all held-out data.

To quantify reinstatement effects across each network, we applied models trained to predict patterns of brain activity during word encoding to epochs just prior to recall (from 900 ms to 100 ms before vocalization onset). For each sample within this window, we computed the ability of each model to decode either the category or serial position of recalled words in the validation data. The resulting AUC time series were smoothed with a 7 ms FWHM Gaussian kernel prior to statistical analysis for noise reduction.

**Statistics and reproducibility**. One-tailed tests were used to assess differences versus chance performance in evaluating decoding accuracy and recall organization. Theoretical chance levels (e.g., an AUC of 0.5 one class vs. all others) were further tested by constructing null distributions via permutation of class labels (i.e., category or serial position).

To evaluate the effects of the number of features and potential epileptiform activity on decoding performance, we modeled classifier performance using linear mixed-effects models. Fixed effects included the number of electrodes sampled, and whether the electrode was localized to the seizure onset zone. Intercepts were allowed to vary, treating the subject as a random variable. Inference was performed through model comparison using a likelihood ratio test, dropping the effect of interest. Model fits and normality of residuals were confirmed through visual inspection.

We corrected for multiple comparisons (across time and frequencies) using a nonparametric permutation procedure. In our analysis of intrinsic connectivity, we performed a nonparametric one-sample $t$ test by constructing a null distribution of the maximum $t$ statistic across frequencies, assuming no difference in intrinsic connectivity for connections within and between networks. This assumption was satisfied by random sign flipping of observed values at the subject level. The significance of observed differences in connectivity strength within versus between networks was compared to a distribution constructed from 2000 random permutations, yielding two-tailed significance with $P_{FWE} < 0.05$.

Multiple comparison correction for network reinstatement followed a similar procedure. We identified significant ($P_{FWE} < 0.05$) clusters in the moments leading up to recall using threshold-free cluster enhancement (TFCE)[75]. The TCFE statistic was computed by taking the original test statistics over the pre-recall period and adjusting by weighting by the height ($h$) and cluster extent ($e$):

$$\sum_k h^H e(h_k)^E dh, \tag{1}$$

where $h_k$ is one of $k$ cluster forming thresholds ($h_k = h_0, h + dh, \ldots, h_{max}$). Height ($H$) and extent ($E$) exponents were set to 2 and 0.5 respectively, with the step size in the cluster threshold ($dh$) set to 0.01. After computing TCFE statistics, we used nonparametric $t$ tests to identify significant clusters based on null distributions ($n = 10000$).

Given the limited availability of iEEG data, no statistical methods were used to predetermine sample sizes (number of subjects). The number of trials included in the experiment was determined by practical constraints of patient testing at epilepsy monitoring units. The subjects in this study were a subset of those included in a multi-site collaboration to investigate the modulation of human memory via direct electrical stimulation. Subjects with at least 5 recording sites located within both the PM and AT networks who performed categorized free recall were included in the present study. All experiments and analyses were performed once, without replication in independent cohorts.

**Reporting summary**. Further information on research design is available in the Nature Research Reporting Summary linked to this article.

## Data availability

De-identified data are available at http://memory.psych.upenn.edu/ Electrophysiological_Data. Source data are provided with this paper.

## Code availability

Analysis code for model fitting and evaluation are available at http://memory.psych. upenn.edu/files/pubs/KragEtal21.code.tgz.

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

## Acknowledgements

We thank Medtronic and Blackrock Microsystems for providing neural recording equipment. This work was supported by the DARPA Restoring Active Memory (RAM) program (Cooperative Agreement N66001-14-2-4032) and NIH grants MH55687 and T32NS047987. We are indebted to all patients who have selflessly volunteered their time to participate in our study. The views, opinions, and/or findings contained in this material are those of the authors and should not be interpreted as representing the official views or policies of the Department of Defense or the U.S. Government. Data were provided in part by the Human Connectome Project, WU-Minn Consortium (Principal Investigators: David Van Essen and Kamil Ugurbil; 1U54MH091657) funded by the 16 NIH Institutes and Centers that support the NIH Blueprint for Neuroscience Research; and by the McDonnell Center for Systems Neuroscience at Washington University. We thank Ethan Solomon and Nora Herweg for providing feedback on this work.

## Author contributions

M.J.K. and Y.E. designed the study; J.E.K. analyzed data and drafted the manuscript. J.E.K. and M.J.K. edited the manuscript. J.M.S. performed anatomical localization of depth electrodes. M.R.S., G.W., B.L., R.G., B.C.J., K.A.Z. and S.A.S. recruited subjects and performed clinical duties associated with data collection.

## Competing interests

M.R.S. is the principal investigator (PI) of research contracts with SK Life Science, Takeda, UCB Pharma, Neurelis, Eisai, Medtronic, Engage, and Cavion. M.R.S. serves as a consultant to Medtronic through Jefferson University which receives all compensation for these services. B.C.J. has research contracts with Neuropace, Medtronic, Sunovion, and Eisai. R.E.G. servers as a consultant to Medtronic, a subcontractor on the RAM project. R.E.G. receives compensation for these services; the terms of this arrangement have been reviewed and approved by Emory University in accordance with its conflict of interest policies. The remaining authors declare no competing interests.
