## [Peer Review File · Nature Communications]

Reviewers' comments:

Reviewer #1 (Remarks to the Author):

Kragel et al investigated reinstatement of temporal and semantic information during memory recall. Using iEEG data from a word-list paradigm the authors first identified two anatomically different functionally connected networks; the AT network which should mainly code for content, and a PM network which should mainly code for serial position. Computational models were trained on the frequency transformed data from the nodes of these networks to encode temporal position or semantic content. Both networks appear to code for both semantic content and serial order. Applying the models to the recall data revealed that the PM network exclusively codes for reinstatement of serial position. The AT network on the other hand seems to code for both serial position and content, however, it appears a bit stronger for content. Finally, an across subject correlation revealed that the individual bias to retrieve either temporally clustered or semantically clustered was correlated with the reinstatement of temporal or semantic information in the PM or AT network, respectively.

General statement: This is certainly an interesting manuscript that reports an exciting new idea. The question as to how time and content is coded in the brain, and what role these two codes play in memory is of utmost interest. This paper could be an important contribution to this area. However, I think there is quite a fundamental flaw in the way the experiment was designed which undermines all results. I also have some concerns as to whether the conclusions drawn by the authors are actually supported by the data.

Major concerns

1. AT and PM networks code semantic and temporal information, respectively. In a number of places the authors suggest that content and temporal context is coded in these two different networks, but I don't think their data is actually showing that. Both networks code for both, serial position and semantic content during encoding. During retrieval, reactivation of category information is present in both networks, albeit somewhat stronger in the AT network. The only data point that is suggestive for a (single) dissociation is the reactivation of serial position which is significantly above chance for the PM network only (and also stronger for the PM compared to the AT). Together it appears that only 1 out of 4 data points suggests that the two networks code for different things.
2. The sample is big yet the effect size appears very small. The AUCs for the retrieval data are barely above chance and presumably only get significant because of the large sample size. This raises the question as to how strong the effects really are.
3. Experimental design. If I understood correctly, then two words of the same category always occurred in sequence during a list (e.g. say you have category A, B, and C, then a possible list sequence was [A, A, C, C, B, B, B, A, A, C, C]). If that is correct then there is an obvious confound between category and list position. In an ideal experimental design that allows to dissociate semantic content from serial position one would construct the lists such that position and content are orthogonal to each other and not conflated like here. I think this is a pretty fundamental issue which undermines the results on all levels (i.e. behaviourally and neutrally). This might also be the reason for why no dissociation was found

between the AT and PM networks in coding for the two dimensions.

Minor comments

4. Classifier analysis. I did not understand how the classification was done from the way it was described on pages 10 and 11. Specifically, I struggled to understand how the authors arrived at a binary classifier when the classifier was trained to distinguish between 12 positions and 25 categories. I must have read the paragraph (lines 215-226) 10 times but could not understand how, from models which predict either 25 categories or 12 positions, binary outcomes could be derived.

Reviewer #2 (Remarks to the Author):

In their manuscript entitled „Distinct cortical systems reinstate content and context information during memory search“, Kragel and colleagues describe results from an intracranial EEG study on the reinstatement of semantic content and temporal context during memory retrieval (measured via free recall). They used a method that had been applied before in fMRI studies to ***

In general, the study addresses an important question – how reinstatement of specific information drives memory retrieval – using a novel and sophisticated method. The authors have recruited a large number of epilepsy patients.

However, the style the paper is written makes it hard to understand what exactly was tested. Many methodological steps remain unclear and should be described in much greater detail. As a result of this lack of clarity, I cannot really evaluate whether the authors' conclusions are justified or not.

Introduction

The description of the background, method, and hypotheses in the introduction should be substantially extended and specified. It seems as if the manuscript was originally prepared as a Brief Communication and not re-written to match the longer format of Nature Communications.

For example, the authors should describe the different methods of the previous studies in greater detail (microelectrode recordings, iEEG studies, fMRI) rather than simply referring to “brain activity”.

The authors state that they aimed to “examine the contributions of cortico-hippocampal networks to memory recall”, but I could not find whether the hippocampus is indeed part of these networks or not. Please clarify.

Results

More detail about the behavioral performance should be provided. Did performance differ between the different categories?

How exactly was figure 1b computed – in particular, how did the authors map the multiple pair-wise similarities between categories onto a 2D space?

Line 51: The theta band is defined as 3-10Hz without any justification. Previous studies in rodents and humans showed different physiology and functional relevance of theta in low and high frequency (i.e., delta/theta vs. theta/alpha). The authors should demonstrate that theta indeed shows a statistically

significant peak in the power spectrum in each individual patient and extract the oscillations frequency at that peak. Similar for low beta (line 52), where even an inference statistics value is missing.

Line 53: I don't understand why "fluctuations in neural activity are correlated across performance of the recall task" – how were the fluctuations in neural activity correlated to performance? And why is this "consistent with their description as intrinsic cortical networks."? Please add a reference that links performance to intrinsic cortical networks.

In line 56, the authors are referring to "oscillatory signals from 3 to 180 Hz". Please indicate how you assessed that there were indeed oscillations across this entire spectrum rather than broadband non-oscillatory activities.

In Figures 2c and 2d, does decoding performance differ between the 2 networks? This does not seem to be the case.

Discussion

In line 115, the authors write "Taken together, our findings argue for an alternative account regarding the neural basis of memory search." – alternative to what?

Do the authors really state that temporal proximity during encoding does not influence recall (lines 118-119), in contrast with several of their previous studies (some of which were based on the same data)?

Methods

The paragraph „Behavioral analysis“ should be extended. How does the distribution of possible temporal lags look like? It would be helpful to have some examples about temporal and categorical clustering values.

Intracranial EEG recordings from the seizure onset zone should be excluded, because it cannot be assumed to measure physiologically and cognitively normal processes. This clinical information should be obtained from all contributing centers and described in a table for all patients (maybe in the Supplement), together with information about MRI lesions and medication at the time of recordings. In addition, the number and distribution of electrodes in all patients should be shown.

How were the resting state recordings conducted – I assume based on pre-implantation fMRI resting state recordings? Which sequence was used? Which scanner? How were these recordings homogenized across sites? Did the authors calculate connectivity in native space or in MNI space? The description of the functional localizers should also be extended. Which kind of partial correlation analysis was applied for the AT network (line 178), and why? Why was apparently a different approach applied to the AT and the PM network?

For the functional connectivity analysis, how did the strength of the connectivity change with increasing distance? Which exact measure of functional connectivity was used? How were the data pre-processed? Please provide more data and a figure.

In line 192/193, the authors write „Networks with greater electrode coverage are likely to have higher decoding accuracy due to the number of features alone.“ – is this really the case? Please show the results. In general if too many irrelevant features are included, decoding approaches may not generalize during cross-validation because of overfitting of data to the training sample.

Wavelet transformation: The authors write that they conducted time-frequency analysis „from the

onset to the offset of stimulus presentation“. How could they then assess power during the ITI?

Regarding the semantic model: The authors write that „Category features were computed by averaging semantic representations across all words presented from a given category.“ Have they also tried to model semantic relatedness between all different exemplars within all categories, rather than between the semantic representations of the average representations within each category? This may allow assessing the variance more accurately.

Regarding the temporal model: How exactly were positions across lists (and sessions) compared to positions within a list? Did they consider the absolute time differences (e.g. in seconds)?

*
Comments by Reviewer #1

1. AT and PM networks code semantic and temporal information, respectively. In a number of places the authors suggest that content and temporal context is coded in these two different networks, but I don't think their data is actually showing that. Both networks code for both, serial position and semantic content during encoding. During retrieval, reactivation of category information is present in both networks, albeit somewhat stronger in the AT network. The only data point that is suggestive for a (single) dissociation is the reactivation of serial position which is significantly above chance for the PM network only (and also stronger for the PM compared to the AT). Together it appears that only 1 out of 4 data points suggests that the two networks code for different things.

The reviewer is correct that semantic and temporal information are present in both networks during encoding. However, we disagree that only 1 out of 4 data points suggest a dissociation between these two networks. While there are small differences in classifier performance during encoding, we show a crossover interaction between classifier (content/context) and network (AT/PM) during recall. Finally, differences in reinstatement between these two networks predict whether memory is organized by either semantic or temporal information, across subjects. As such, all data points evaluated during retrieval (4 out of 4) suggest that distinct forms of information are reinstated across these two networks.

We agree with the reviewer that the language in the previous version of the manuscript emphasized distinct coding across these two networks. We have revised the introduction (p. 3, lines 24-35) to discuss the representation of episodic content within the PM network, and the discussion (p. 10,

lines 188-193) to emphasize that this distinction only occurs during retrieval. In addition, we provide additional discussion of neural coding within each network based on previous studies examining memory reinstatement. Most theories of PM network function suggest that it represents retrieved episodic memories (including content and spatiotemporal context). These theories are consistent with our results, wherein both types of information were present during retrieval. We believe that our revised manuscript now clearly communicates the main contribution of this work, which is the dissociation during retrieval and relationship with behavior.

2. The sample is big yet the effect size appears very small. The AUCs for the retrieval data are barely above chance and presumably only get significant because of the large sample size. This raises the question as to how strong the effects really are.

The effect sizes from our reinstatement analysis are small; however, these effect sizes are likely underestimated as our analysis approach required us to match the number of features/electrodes across the two networks. We now include a supplemental analysis showing how classification performance varies with the number of electrodes sampled. As can be seen in Supplementary Figure 4, we generally find linear increases in classification performance with increasing features (with the exception of category decoding from the PM network during encoding, which remains fairly stable with increasing features). These results suggest that semantic and temporal information is represented in a distributed manner across these networks, and implies that greater or higher resolution sampling of a given network would lead to larger effects. To make sure readers are aware of this point, we describe this analysis when reporting the results (p. 8, lines 138-147).

3. Experimental design. If I understood correctly, then two words of the same category always occurred in sequence during a list (e.g. say you have category A, B, and C, then a possible list sequence was [A, A, C, C, B, B, B, B, A, A, C, C]). If that is correct then there is an obvious confound between category and list position. In an ideal experimental design that allows to dissociate semantic content from serial position one would construct the lists such that position and content are orthogonal to each other and not conflated like here. I think this is a pretty fundamental issue which undermines the results on all levels (i.e. behaviourally and neutrally). This might also be the reason for why no dissociation was found between the AT and PM networks in coding for the two dimensions.

The reviewer is correct in their understanding of the list structure used in the experiment (with the exception that no more than two same-category pairs were presented consecutively). However, we do not believe the list structure confounds the interpretation of our results. In the present work, there are two potential measures that could be confounded by list structure: 1) decoding performance for serial positions and categories, and 2) behavioral measures of recall organization.

Regarding classification performance, we trained models to learn the category and serial position

of items sampled from *different lists*. Because the assignment of categories to serial positions was randomized across lists, the position of an item would not provide diagnostic information about the category of a given item (that is, they are orthogonal). In addition, same-category items were always presented in pairs, with the category always changing after each pair. As a result, shifts between categories were balanced across the experiment. It is worth pointing out that it is impossible for semantic and temporal information to be orthogonal *within* a list, as the sequence of semantic information can be used to define a temporal context (e.g., the list started with a fruit). For these reasons, we are confident that the list structure does not undermine our ability to identify changes in electrophysiology related to the content or context of items in memory.

Regarding behavioral measures of recall organization, it is possible that our estimates of temporal and categorical organization were biased due to the list structure. To rule out this possibility, we now report a simulation-based control analysis to verify our measures of recall organization. To examine the influence of category pairs on temporal organization, we generated random recall sequences that were matched to the observed level of categorical organization an number of items recalled for each list. Then, we computed a null distribution ($n = 1000$) of temporal factor scores from these recall sequences. This distribution reflects chance levels of temporal organization, controlling for observed levels of category organization and the confounding list structure. We standardized observed measures of temporal organization, using the mean and standard deviation of these null distributions. These standardized measures indicate the amount of temporal organization one would expect from chance if the observed recall sequences were generated without temporal factors influencing recall organization while controlling for the list structure. We also computed a similar measure of category clustering that controls for the list structure. We now report this control analysis on page 5 of the manuscript (p. 4, lines 64-68), and depict the results in Supplementary Figure 2. In brief, these simulation analyses show that there is evidence for both temporal and categorical organization within our experiment. Further, measures of across subject variability were consistent for both measures of temporal organization ($r_{67} = 0.92$, $p < 0.0001$) and category organization ($r_{67} = 0.56$, $p = 0.0001$).

*

Minor Comments by Reviewer 1

4. Classifier analysis. I did not understand how the classification was done from the way it was described on pages 10 and 11. Specifically, I struggled to understand how the authors arrived at a binary classifier when the classifier was trained to distinguish between 12 positions and 25 categories. I must have read the paragraph (lines 215-226) 10 times but could not understand how, from models which predict either 25 categories or 12 positions, binary outcomes could be derived.

We apologize for the lack of clarity in how classification was done. We have clarified our description of this process in the Methods. In brief, our models estimate neural states (patterns of spectral power across frequencies and electrodes) for a given category or serial position. These estimated neural states are correlated with the observed pattern of neural activity during encoding (for model validation, Fig. 2c,d) or retrieval (for reinstatement, Fig. 3b). The similarity of the observed

neural activity to each class was then normalized to a probability using a softmax operation, giving an estimate that a neural state reflected a specific serial position or category. From these probabilities, classifier performance for each class was estimated using a generalization of AUC to the multidimensional case (Hand and Till, 2001). Classifier performance was then *averaged across classes to provide an aggregate AUC measure*.

*
Comments by Reviewer #2

1. The description of the background, method, and hypotheses in the introduction should be substantially extended and specified. It seems as if the manuscript was originally prepared as a Brief Communication and not re-written to match the longer format of Nature Communications. For example, the authors should describe the different methods of the previous studies in greater detail (microelectrode recordings, iEEG studies, fMRI) rather than simply referring to "brain activity".

The reviewer is correct that the manuscript was originally prepared for a shorter format. We now provide additional details within the introduction, to better frame our study (pages 3-4, lines 24-35).

2. The authors state that they aimed to "examine the contributions of cortico-hippocampal networks to memory recall", but I could not find whether the hippocampus is indeed part of these networks or not. Please clarify.

We now provide additional detail regarding the individual parcels within each network of interest as well as sampling across subjects in Supplementary Figure 1.

0. More detail about the behavioral performance should be provided. Did performance differ between the different categories?

We now describe differences in recall performance with category and serial position (p. 4, lines 48-54). We now display these results in Figure 1.

0. How exactly was figure 1b computed - in particular, how did the authors map the multiple pair-wise similarities between categories onto a 2D space?

Figure 1b reduced the word2vec feature space (300 dimensions) to a 3-dimensional space using principal component analysis. We now explain this in the text (p. 5, lines 60-62).

5. Line 51: The theta band is defined as 3-10Hz without any justification. Previous studies in rodents and humans showed different physiology and functional relevance of theta in low and high frequency (i.e., delta/theta vs. theta/alpha). The authors should demonstrate that theta indeed shows a statistically significant peak in the power spectrum in each individual patient and extract the oscillations frequency at that peak. Similar for low beta (line 52), where even an inference statistics value is missing.

We were not attempting to define the theta band with this statement, but rather highlight that the frequencies that demonstrated increased coupling within this network were consistent with the theta band. The purpose of this analysis was not to identify differences in coupling between theta or beta oscillations between the two networks, but rather as a general test of whether the features we used in our modeling approach were more consistent within as opposed to between the two networks. We believe an in-depth characterization of narrowband oscillatory peaks, their differences across cortical systems, and their involvement in representing the contents of memory is beyond the scope of the present work.

For the reviewer, we report an additional analysis that models observed power spectra as a combination of periodic (oscillatory) and aperiodic components. After whitening power spectra (i.e., removing aperiodic components), we find that oscillatory activity contributes significantly to changes in spectral power during task performance (panel a) within both theta and beta bands. In addition, we observed network-level differences in oscillatory power that were consistent with our connectivity results (panel b), with increased oscillatory power at lower frequencies within the AT network (see Supplementary Fig. 4). These findings indicate that the present results reflect a mixture of both oscillatory and broadband effects, as indicated in the manuscript. We highlight this point in the discussion (p. 10, lines 194-204).

6. Line 53: I don't understand why "fluctuations in neural activity are correlated across performance of the recall task" - how were the fluctuations in neural activity correlated to performance? And why is this "consistent with their description as intrinsic cortical networks."? Please add a reference that links performance to intrinsic cortical networks.

We apologize for a lack of clarity in this sentence. We have rephrased this sentence to emphasize our concluding point: These results show that task-related changes in spectral power are similar within these two networks, consistent with their description as dissociable networks from resting-state fMRI (p. 6, lines 100-101). We have also expanded our motivation for this analysis (pages 5-6, lines 78-87) to better justify this interpretation.

7. In line 56, the authors are referring to "oscillatory signals from 3 to 180 Hz". Please indicate how you assessed that there were indeed oscillations across this entire spectrum rather than broadband non-oscillatory activities.

As mentioned in our response to point 6, we now avoid using the term oscillatory signals when referring to changes in spectral power.

3. In Figures 2c and 2d, does decoding performance differ between the 2 networks? This does not seem to be the case.

We now report additional tests of decoding performance during encoding. We find a small but significant increase in the ability to decode serial position from the PM vs. AT network; these comparisons are now reported in the text (p. 7, lines 116-119).

0. In line 115, the authors write "Taken together, our findings argue for an alternative account regarding the neural basis of memory search." - alternative to what?

We have clarified this sentence to communicate that we are arguing against activity in a single cortical region (e.g., lateral prefrontal cortex) driving hippocampal-dependent recall (p. 10, lines 205-207).

0. Do the authors really state that temporal proximity during encoding does not influence recall (lines 118-119), in contrast with several of their previous studies (some of which were based on the same data)?

We were not claiming that temporal proximity does not influence recall; rather, we were claiming that it is possible to use retrieval cues that do not contain temporal information. For example, representations of semantic content in the AT system can target items from a particular category, without driving temporal reinstatement. We have expanded the discussion (pages 10-11, lines 205-216) to clarify this point, and still claim that temporal reinstatement is essential to targeting memories in from a specific episodic context.

1. The paragraph "Behavioral analysis" should be extended. How does the distribution of possible temporal lags look like? It would be helpful to have some examples about temporal and categorical clustering values.

The distribution of possible lags decreases linearly for longer lags, with a bias in the forward direction (see the attached figure). This results from the tendency of subjects to initiate recall at the beginning

of the list (primacy), which limits possible backward transitions. We do not feel this level of detail is useful for the average reader. However, we have modified Figure 1 to better illustrate what different temporal and categorical factor scores reflect, in terms of recall sequences on the task (Fig. 1e).

12. Intracranial EEG recordings from the seizure onset zone should be excluded, because it cannot be assumed to measure physiologically and cognitively normal processes. This clinical information should be obtained from all contributing centers and described in a table for all patients (maybe in the Supplement), together with information about MRI lesions and medication at the time of recordings. In addition, the number and distribution of electrodes in all patients should be shown.

To address the reviewer's concern, we have included Table S1 that provides descriptive information regarding patient demographics and relevant clinical information. We now depict electrode coverage across all patients in Supplementary Figure 1, along with coverage in either the PM or AT networks. We perform an additional analysis showing that including electrodes from the seizure onset zone has no impact on the findings of our analysis (p. 8, lines 148-153, Supplementary Fig. 5).

13. How were the resting state recordings conducted - I assume based on pre-implantation fMRI resting state recordings? Which sequence was used? Which scanner? How were these recordings homogenized across sites? Did the authors calculate connectivity in native space or in MNI space? The description of the functional localizers should also be extended.

We are sorry for the lack of clarity in the previous version of the manuscript. Networks defined from resting state fMRI were defined based on data from the Human Connectome Project, not resting state scans collected from individual patients. Additionally, no functional localizers were used in the present study. We have reworded the description of the fMRI data on page 5, line 71, to provide

additional clarity.

14. Which kind of partial correlation analysis was applied for the AT network (line 178), and why? Why was apparently a different approach applied to the AT and the PM network?

The same measure of connectivity (the correlation between a parcel and the seed parcel, covarying out signal from all other parcels) was used for each network. We now clarify this in the methods (p. 13, lines 284-289).

15. For the functional connectivity analysis, how did the strength of the connectivity change with increasing distance? Which exact measure of functional connectivity was used? How were the data pre-processed? Please provide more data and a figure.

As presented in Glasser et al. 2016, one can construct whole-brain networks many different measures of connectivity. We used measures of linear partial correlation between each cortical parcel and the seed (i.e., either anterior temporal lobe or posterior parietal cortex) in order to exclude regions that may reflect indirect connectivity. Notably, without this approach, nodes have high connectivity to a number of regions not functionally implicated in the representation of context and content. We now characterize how functional connectivity (in our iEEG data) changes with distance in Supplementary Fig. 3, which shows increased functional connectivity at proximal recording sites.

16. In line 192/193, the authors write "Networks with greater electrode coverage are likely to have higher decoding accuracy due to the number of features alone." - is this really the case? Please show the results. In general if too many irrelevant features are included, decoding approaches may not generalize during cross-validation because of overfitting of data to the training sample.

We agree with the reviewer that including a high number of features can lead to overfitting. This is the primary reason we used regularized regression in developing our models. As the number of features included in our decoding analyses have the potential to influence classifier performance (see comment 2 to Reviewer 1), we have now included additional analyses (Supplementary Figures 4 and 5) that demonstrate an increase in decoding accuracy with the number of electrodes (features).

17. Wavelet transformation: The authors write that they conducted time-frequency analysis "from the onset to the offset of stimulus presentation". How could they then assess power during the ITI?

We did not analyze any signal during the ITI.

18. Regarding the semantic model: The authors write that "Category features were computed by averaging semantic representations across all words presented from a given category." Have they also tried to model semantic relatedness between all different exemplars within all categories, rather than between the semantic representations of the average representations within each category? This may allow assessing the variance more accurately.

In an additional analysis of this dataset, we modeled the semantic representation of each item as suggested by the reviewer. Notably, there was no difference in decoding performance during either encoding or retrieval when comparing semantic information at the item or category level. This is likely due to the manner in which items were selected, in which they were commonly generated exemplars from each category. Thus, differences between categories dominate the semantic space across all items. We elected not to include this analysis as it would detract from the focus of the manuscript – dissociating the reinstatement of content and context representations during memory search.

19. Regarding the temporal model: How exactly were positions across lists (and sessions) compared to positions within a list? Did they consider the absolute time differences (e.g. in seconds)?

In evaluating classifier performance, all serial positions were considered irrespective of where they occurred within each session, and on which session they occurred.

REVIEWERS' COMMENTS

Reviewer #1 (Remarks to the Author):

General assessment:

The authors have addressed some of the concerns raised in the previous round, however, the major concern still remains. This concern is that the conclusions are not fully justified by the data, because a dissociation in the coding of semantic and temporal appears to not be strongly present in the data. Therefore I am not convinced that this manuscript is making a convincing case.

Specific concern:

In my previous review I remarked that the evidence for a difference in coding of semantic content and temporal context between the two networks appears to be weak. This is because at encoding both networks decode for both dimensions, and the difference between the two networks barely is significant ($p < 0.04$; which in conjunction with the large sample size indicates a very small effect size). In their response, the authors highlight the significant cross-over interaction, but such an interaction is not sufficient to support a double-dissociation of coding in the two networks, because that interaction can be driven by only one data point. The data at retrieval seems a bit more convincing in terms of coding for temporal context, but content coding again is present in both networks. Together, I am not convinced that the data presented here shows evidence that content and temporal context is coded in two different networks.

Reviewer #2 (Remarks to the Author):

The authors provided a thorough revision and substantially modified and extended the manuscript. All of my previous comments have been addressed.

*Comments by Reviewer #1

General assessment:

The authors have addressed some of the concerns raised in the previous round, however, the major concern still remains. This concern is that the conclusions are not fully justified by the data, because a dissociation in the coding of semantic and temporal appears to not be strongly present in the data. Therefore I am not convinced that this manuscript is making a convincing case.

Specific concern:

In my previous review I remarked that the evidence for a difference in coding of semantic content and temporal context between the two networks appears to be weak. This is because at encoding both networks decode for both dimensions, and the difference between the two networks barely is significant ($p < 0.04$; which in conjunction with the large sample size indicates a very small effect size). In their response, the authors highlight the significant cross-over interaction, but such an interaction is not sufficient to support a double-dissociation of coding in the two networks, because that interaction can be driven by only one data point. The data at retrieval seems a bit more convincing in terms of coding for temporal context, but content coding again is present in both networks. Together, I am not convinced that the data presented here shows evidence that content and temporal context is coded in two different networks.

We agree with Reviewer 1 that our data do not provide strong evidence for a dissociation between the two networks at encoding. However, our main hypothesis regards reinstatement of content and context during retrieval. We show greater reinstatement of content in the AT network, greater reinstatement of context in the PM network, and an interaction between the two. Thus, we believe these data justify our claims. We have carefully proofed our manuscript to make sure we interpret our findings with respect to reinstatement, and do not claim any dissociation between the two networks at encoding. Further, we have revised the abstract and now specifically report that both content and context information can be decoded from both networks at encoding.

*Comments by Reviewer #2

The authors provided a thorough revision and substantially modified and extended the manuscript. All of my previous comments have been addressed.

We thank the reviewer for their feedback.